# Peritoneal Seeding Is More Common in Gastric Cancer Patients with FGFR2 Amplification or High Tumor Mutation Burden

**DOI:** 10.3390/diagnostics12102355

**Published:** 2022-09-28

**Authors:** Hyunjin Kim, Sujin Park, So Young Kang, Soomin Ahn, Kyoung-Mee Kim

**Affiliations:** 1Department of Pathology and Translational Genomics, Samsung Medical Center, Sungkyunkwan University School of Medicine, Seoul 06351, Korea; 2Pathology Center, Seegene Medical Foundation, Seoul 06351, Korea; 3Center of Companion Diagnostics, Samsung Medical Center, Seoul 06351, Korea

**Keywords:** fibroblast growth factor receptor-2, gastric carcinoma, peritoneal seeding

## Abstract

Fibroblast growth factor receptor-2 (FGFR2) gene alterations have been identified in solid tumors. FGFR2 amplification is found in 2–9% of gastric carcinomas. We hypothesized that FGFR2 could be associated with peritoneal seeding and studied 360 advanced gastric carcinoma patients; 222 (61.7%) were male, 246 (73.7%) had poorly differentiated histology, and 175 (48.6%) presented with peritoneal seeding. High tumor mutation burden (TMB) was observed in 44 (12.2%) patients, high microsatellite instability (MSI) was observed in 12 (3.33%) patients, ERBB2 amplification was observed in 44 (12.2%) patients, EBV positivity was observed in 10 (10/278; 3.6%) patients, and PD-L1 positivity was observed in 186 (186/264; 70.5%) cases. We found FGFR2 amplification in 26 (7.2%) patients, of which 12 (46.2%) were female and 22 (84.6%) had poorly differentiated histology. In these 26 cases, the copy number of FGFR2 amplification ranged from 3.7 to 274. Eighteen of them showed seeding, and this association was statistically significant (18/26, 69.2%; 157/334, 47%; *p* = 0.023). In addition, high TMB was significantly associated with seeding (*p* = 0.028; OR = 1.83). Poorly differentiated histology was significantly associated with seeding (*p* = 0.04) but not with FGFR2 amplification (*p* > 0.1). Seeding was frequent in gastric carcinoma patients with FGFR2 amplification, in patients with high TMB, or in those who were female. The subgroup of patients with FGFR2 amplification could be potential candidates for targeted therapeutic agents.

## 1. Introduction

Gastric carcinoma (GC) is the fourth most common cancer and the third deadliest cancer worldwide, according to GLOBOCAN 2020 data [1]. While adjuvant or neoadjuvant chemotherapy has shown prognostic benefits in advanced GC, the overall survival remains poor [2,3]. Of all the modes of systemic spread of GC, peritoneal seeding is the most frequent and most disastrous [4]. This contrasts with other carcinomas that spread through lymphatic channels or blood vessels. According to Sarela et al., the 5-year survival rate of unresectable metastatic GC is less than 5% [5,6].

The mechanism of peritoneal seeding has been widely studied but has yet to be determined and characterized. The epithelial–mesenchymal transition (EMT) and associated chemokines have been investigated [7]. Others have proposed that hypoxia-inducible factor-1a mediates seeding by vascular networks [8]. More historically, in 1993, Shimotsuma et al. proposed that capillary lymphatic vessels of the omentum are the primary source of dispersion for tumor cells [9].

Since GC with peritoneal seeding is particularly incurable, several treatment modalities, such as cytoreductive surgery and heated intraperitoneal chemotherapy, have been under clinical trials. However, the treatment score has not been proven to be superior to conventional systemic chemotherapy [10].

In our routine clinical practice, we performed next-generation sequencing (NGS) with TruSight Oncology (TSO) 500 in advanced GC patients to identify a molecular target for treatment in palliative settings. We found a trend that FGFR2 alterations are frequently observed in patients with peritoneal seeding. The fibroblast growth factor receptor (FGFR) is a protein consisting of four transmembrane receptors that translates downstream signals for cell proliferation, migration, differentiation, and angiogenesis [11]. This protein is frequently altered in urothelial and intrahepatic cholangiocarcinomas [12]. In 2020, FGFR-targeting erdafitinib and pemigatinib were approved by the Food and Drug Administration for use in FGFR3-mutated urothelial carcinomas and FGFR2-fusion cholangiocarcinomas [13]. Among several types of FGFR alterations, FGFR2 amplification has been identified in endometrial and gastric cancers (GC) and is associated with poor prognosis [14]. In GC, the incidence of FGFR2 amplification has been reported in 2–9% of cases and more frequently in diffuse histological types [14,15]. Moreover, FGFR2 amplification is associated with resistance to chemotherapy and sensitivity to FGFR-targeting tyrosine kinase inhibitors [16].

In our study, we hypothesized that FGFR2 amplification could be associated with peritoneal seeding and may be an important therapeutic target in this dismal patient subgroup.

## 2. Materials and Methods

We recorded and reviewed the results of mass parallel sequencing in 360 patients with advanced GC (stage III and IV). All patients received systemic chemotherapy at the Samsung Medical Center from October 2019 to June 2021. Genomic profiling was performed for formalin-fixed paraffin-embedded (FFPE) blocks using a TSO 500 panel and bioinformatics pipeline (Illumina, San Diego, CA, USA). A gene copy number greater than two was defined as amplification.

Programmed death-ligand 1 (PD-L1) immunohistochemistry (IHC) was performed with the PD-L1 22C3 pharmDx assay (Agilent Technologies, San Diego, CA, USA) in 3 μm thick tissue sections of the 264 available FFPE blocks made from biopsy or resected specimens. The pharmDx 22C3 assay was stained with the EnVision FLEX visualization system (Agilent, Santa Clara, CA, USA) on an Autostainer Link 48 system (Dako), along with positive and negative controls, according to the manufacturer’s instructions. Two gastrointestinal pathologists (S.A. and K.-M.K.) evaluated the combined positive score (CPS) of the membranous staining of immune cells and tumor cells, as described in previous studies [17]. CPS ≥ 1% was considered PD-L1-positive. For FGFR2, IHC was performed with the anti-FGFR2 primary antibody (EPR5180) (1:150; Abcam, Cambridge, UK) on 3 μm thick tissue sections. Representative FFPE blocks from four surgically resected cases were selected for a preliminary study. After deparaffinization and rehydration, antigen retrieval and peroxidase blocking were performed. The sections were then incubated with primary antibodies. The BOND-MAX fully automated IHC staining system (Leica) was used according to the manufacturer’s protocol. For interpretation, strong cytoplasmic and membranous staining was interpreted as positive (overexpression).

EBV-encoded RNA in situ hybridization was performed on 3 μm thick FFPE-block sections in 278 available cases with BOND-MAX autoimmunostainer (Leica Biosystems, Melbourne, Australia) with an EBV-encoded RNA probe (Leica, Newcastle, UK). All steps were performed according to the manufacturer’s instructions.

For TSO500 NGS sequencing, DNA extraction, library preparation, sequencing, and data analysis were performed as described previously [17] and TMB-H was defined as 10 or more mutations per megabase (Mb).

Statistical analyses were performed using SPSS (version 25.0; IBM, Armonk, NY, USA). The electronic medical records of all the patients were reviewed thoroughly. Peritoneal seeding was identified using either radiologic or intraoperative evaluation.

## 3. Results

Of the 360 patients, 222 (61.7%) were male and 138 (38.3%) were female. Of these, 246 (68.3%) had poorly differentiated GC and 114 (31.7%) had well- to moderately differentiated GC. A total of 175 (48.6%) patients presented with peritoneal seeding. We found FGFR2 amplification in 26 (7.2%) patients, of whom 14 (53.8%) were male and 12 (46.2%) were female; 22 (84.6%) had poorly differentiated GC and 4 (15.4%) were well- to moderately differentiated. The copy number of FGFR2 amplification in the 26 cases ranged from 3.7 to 274 (mean 52.7; median 19.4). Of the 360 cases, high tumor mutation burden (TMB) was observed in 44 (12.2%), high microsatellite instability (MSI) was observed in 12 (3.33%) patients, ERBB2 amplification was observe in 44 (12.2%) patients, EBV positive was observed in 10 (10/278; 3.6%) patients, and PD-L1 positivity was observed in 186 (186/264; 70.5%) patients. Three representative images of FGFR2 amplified cases are shown in Figure 1.

Among the 26 FGFR2-amplified cases, 18 (69.2%) showed peritoneal seeding, and this association was statistically significant (18/26, 69.2%; 157/334, 47%; *p* = 0.023). At a 95% confidence interval, the odds ratio (OR) was 2.54 (1.07–6.00). (Table 1) We also explored other genomic signatures to identify alterations associated with peritoneal seeding. Unexpectedly, the high TMB cases were statistically significant with peritoneal seeding (*p* = 0.028), with an OR of 1.83 (1.02–3.30), while microsatellite instability status, ERBB2 amplification, EBV status, and PD-L1 expression were not associated (Fisher’s exact test; *p* = 0.22, 0.057, 0.21, and 0.19, respectively). Although poorly differentiated carcinomas were associated with peritoneal seeding (*p* = 0.04), FGFR2 amplification was not (*p* = 0.158).

Among the 26 FGFR2-amplified cases, FGFR2 fusion was found in eight cases, of which TACC2 was the most frequent fusion partner, present in three cases (among which one had peritoneal seeding). Other fusion partners were WDR11, BTBD, C10orf90, INPP5F, and PPAPDC1A; three of the five cases had peritoneal seeding. In six of the cases with no FGFR2 amplification, one case had FGFR3 amplification and fusion (and peritoneal seeding), two cases had FGFR1 amplification only (one of these had peritoneal seeding), and one had FGFR1 amplification and fusion without peritoneal seeding. One patient had FGFR2 mutation (and peritoneal seeding), and the other had FGFR4 amplification without peritoneal seeding. For the treatment of FGFR2-amplified GC patients, adjuvant or palliative chemotherapy was administered to 25 cases because one patient had refused chemotherapy. Specific regimen includes xelox (capecitabine and oxaliplatin) in all but one of the cases as the first-line chemotherapy. For patients with progress of disease after xelox, paclitaxel (taxol) and ramucirumab (*n* = 12), folfox (oxaliplatin-5-fluorouracil-leucovorin) (*n* = 5) or folfiri (leucovorin calcium, fluorouracil, and irinotecan) (*n* = 2) combination chemotherapies, and pembrolizumab (*n* = 5) were used as the second- or third-line chemotherapy. In one patient, Derazantini, a potent FGFR1–3 kinase inhibitor, was administered. The clinicopathological information of the abovementioned FGFR-altered cases is summarized in Table 2.

Of the 185 patients without peritoneal seeding, 56 (30.3%) were female, while in the peritoneal seeding group with 175 patients, 82 (46.9%) were female; the difference in sex distribution was significant (*p* < 0.001). Due to the short follow-up period, the prognostic significance was not evaluable. In multivariate logistic regression analyses with factors shown as significant in the univariate analysis, female sex (*p* = 0.004) remained significant while FGFR2 amplification (*p* = 0.05) and TMB-high (*p* = 0.06) showed marginal significance.

In summary, peritoneal seeding was significantly associated with being female and having poorly differentiated histology, FGFR2 amplification, and high TMB genomic alterations.

## 4. Discussion

Peritoneal carcinomatosis is the most frequent and lethal mode of GC progression [18]. Recent advances in the modes of cancer treatment have led to intense research on the molecular basis of GC. The molecular basis of peritoneal dissemination has been studied previously; in one study, Kurashige et al. demonstrated that DDR2 overexpression is associated with peritoneal spread in xenograft mouse models [19]. In another group, Lim et al. performed whole-exome sequencing in eight patients with GC and found 24 recurrent mutations [18].

In our routine clinical practice, we performed next-generation sequencing with TSO 500 in advanced GC patients to identify a molecular target in palliative settings. FGFR2 amplification was notable among the recurrent mutations. Among several known FGFR alterations, amplification is more frequent than fusion or mutation. From the FGFR alterations, FGFR2 amplification is one of the less frequent, reported as 0.34% of GC cases in a review article by De Luca et al. [20]. This alteration has been identified in esophageal, gastric, and breast carcinomas [21]. Xiet et al. reported the therapeutic potential of AZD4547, a potent and selective inhibitor of FGFR in patients with FGFR2-amplified GC cell lines SNU-16 and KATOIII [22].

In our study, we explored the association between FGFR2 amplification and peritoneal seeding and found for the first time that peritoneal seeding is significantly associated with FGFR2 amplification and that FGFR2 amplification is 2.54 times more likely to occur in peritoneal seeding.

FGFR2 immunohistochemistry was performed at the research level in some institutions. We performed a pilot study to determine whether a high copy number was associated with a strong staining intensity of tumor cells. Unfortunately, FGFR2 IHC staining results and the copy number of FGFR2-amplified cases were irrelevant. Moreover, the background staining of muscle and other infiltrated cells was intense. Since most of our patients were in palliative settings, among the 26 FGFR2-amplified cases, only four were surgical resection specimens. Other samples included peritoneal biopsies from open and closed surgeries or endoscopic biopsies and were unsuitable for analyzing the IHC staining patterns. For FGFR2 analysis, mass general sequencing such as TSO 500 would be more reliable and efficient than IHC. Ahn et al. used the FGFR2b antibody, which showed high correlation with the FGFR polymerase chain reaction (PCR) results [23]. Due to the shortage of commercially available FGFR2b antibodies, we were not able to perform FGFR2b IHC in our case series.

Among other signatures, high TMB was significantly associated with peritoneal seeding. TMB has been widely studied as a predictive marker of response to immunotherapy [24]. Chen et al. reviewed multiple factors that predict peritoneal dissemination in GC, including female sex, diffuse-type histology, Borrmann’s classification type IV, and venous invasion [25,26,27,28]. Consistent with previous studies, we found a significant association between peritoneal seeding and female sex, and between peritoneal seeding and poorly differentiated histology. However, peritoneal seeding was not associated with microsatellite instability (MSI) status, ERBB2 amplification, or PD-L1 expression (Fisher’s exact test; *p* = 0.22, 0.057, and 0.19, respectively). Moreover, an association between FGFR2 amplification and poorly differentiated carcinomas was not observed.

FGFR alteration has been of great interest to clinicians and researchers due to the targeted drug development. In a large multi-institutional study, 7% of 4853 solid tumors were identified as having some kind of FGFR alteration [29]. All four types of FGFR (FGFR1–4) have been associated with specific cancer types, and FGFR2 alteration has been most widely found in GC. Among the 26 FGFR2-amplified cases, 8 (30.7%) cases were found with FGFR2 fusion.

The most frequent concurrently observed fusion was TACC2-FGFR2. This fusion has been reported in a stage IV pancreatic adenocarcinoma patient with durable, complete response to erdafitinib, which is a pan-FGFR inhibitor [30]. In a multicenter, phase 1–2 study (NCT01752920), 29 FGFR2 fusion-positive inoperable intrahepatic cholangiocarcinoma patients were treated with derazantinib, another pan-FGFR inhibitor, with promising anti-tumor activity and safety profile, although the fusion partners of FGFR2 have not been clearly demonstrated [31]. TACC2-FGFR2 fusion was identified in 0.02% of AACR GENIE cases, in intrahepatic cholangiocarcinomas and adenocarcinomas of the gastrointestinal tract from the esophagus to the colon [21].

FGFR2 fusion was studied more in-depth in advanced cholangiocarcinoma [32]. A phase II clinical trial was performed, where the patients were grouped into one of three cohorts: harboring FGFR2 fusion or rearrangement, harboring other FGF/FGFR alterations, or harboring no FGF-related alterations. Their data supported the efficacy of pemigatinib, a selective inhibitor of FGFR1, 2, and 3, in the FGFR2 fusion/rearrangement group with a centrally confirmed objective response. Another review has shown that FGFR-altered tumors (all cancer types) are more likely to respond to FGFR inhibitors [20].

There have been several studies that showed worse survival of FGFR2-amplified or overexpressed GCs [33,34]. Although a selective FGFR2 inhibitor has been shown to be more sensitive to FGFR2-amplified GC cell-lines, there are four ongoing clinical trials (NCT01719549, NCT01921673, NCT02052778, and NCT02318329) to assess the efficacy in actual patients [22].

Although previous studies have reported FGFR2 amplification in gastric cancers and its association with a poor outcome, our study has shown for the first time that FGFR2 amplification is significantly associated with peritoneal seeding. Further studies with larger sample sizes are warranted to confirm our results. Moreover, a prospective study on FGFR-altered GC cases with respect to targeted drug response would be valuable.

## 5. Conclusions

Peritoneal seeding was frequent in advanced GC patients who had FGFR2 amplification, had high TMB, and were female. The subgroup of patients with FGFR2 amplification could be potential candidates for targeted therapeutic agents.

## Figures and Tables

**Figure 1 diagnostics-12-02355-f001:**
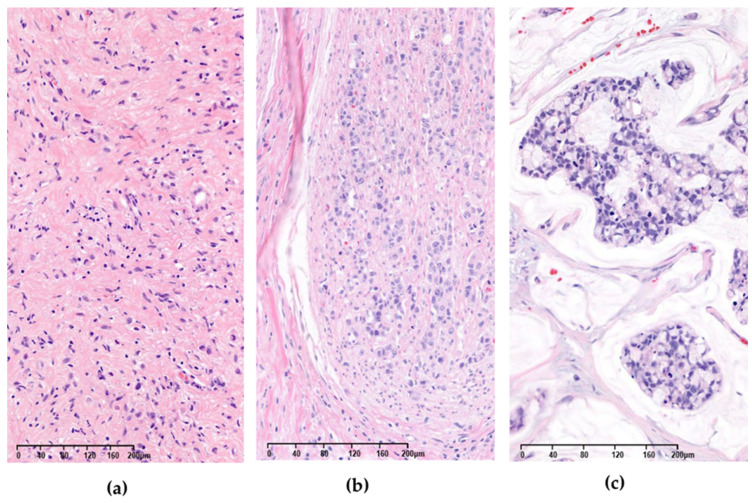
Three different gastric adenocarcinoma cases with FGFR2 amplification (hematoxylin and eosin stain). Copy numbers of each case were (**a**) 9.1 (ID 205); (**b**) 9.2 (ID 253); and (**c**) 64 (ID 242), respectively.

**Table 1 diagnostics-12-02355-t001:** Summary of 360 cases with key features. Associations between peritoneal seeding and FGFR2 amplification; peritoneal seeding and poorly differentiated (P/D) carcinoma were statistically significant. Associations between FGFR2 amplification and P/D carcinoma were not significant.

		FGFR2 Amplification		Total
		Positive	Negative	
**Peritoneal Seeding**	Positive	18 (P/D: 15)	157 (P/D: 123)	175 (P/D: 138)
	Negative	8 (P/D: 7)	177 (P/D: 123)	185 (P/D: 130)
**Total**		26 (P/D: 22)	334 (P/D: 246)	360 (P/D: 268)

**Table 2 diagnostics-12-02355-t002:** Clinicopathological information of 26 patients with FGFR-altered gastric cancer.

ID	SEX	AGE	FGFR2 Amplification	Peritoneal Seeding	FGFR2 Copy Number	Other FGFR Alterations	Differentiation	PD-L1 CPS	ERBB2 Amplification	HIGH TMB	MSI	EBV
**5**	F	42	Y	Y	27.6	FGFR2-WDR11	PD	Neg	N	N	N	Neg
**34**	F	52	Y	Y	57	none	PD	Pos	N	N	N	Neg
**49**	M	53	Y	Y	19.6	FGFR2-TACC2	PD	Pos	N	N	N	Neg
**74**	M	45	Y	Y	254	none	PD	Neg	N	N	N	Neg
**78**	M	52	Y	N	20.8	none	MD	Pos	Y	N	N	Neg
**109**	F	33	Y	Y	6.2	none	PD	ND	N	N	N	ND
**122**	M	28	Y	N	4.3	FGFR2-BTBD	PD	Pos	N	N	N	Neg
**143**	F	50	Y	Y	19.1	none	PD	Pos	N	N	N	Neg
**145**	F	48	Y	Y	15.3	none	MD	ND	N	N	N	ND
**146**	M	76	Y	Y	52	none	MD	Pos	Y	Y	N	ND
**155**	F	70	Y	Y	32.8	none	PD	Neg	N	N	N	Neg
**163**	M	67	Y	N	17.8	FGFR2-TACC2	PD	Pos	N	N	N	Neg
**173**	M	54	Y	Y	93.5	none	PD	ND	N	N	N	Neg
**205**	F	48	Y	Y	9.1	none	PD	Neg	N	N	N	Neg
**208**	M	50	Y	Y	274	FGFR2-C10orf90	PD	ND	N	N	N	ND
**221**	M	54	Y	Y	6.2	none	MD	Neg	N	N	N	ND
**222**	F	58	Y	Y	185.9	FGFR2-INPP5F	PD	Neg	N	N	N	ND
**242**	M	60	Y	N	64.1	FGFR2-PPAPDC1A	PD	Pos	N	N	N	Neg
**251**	M	61	Y	N	66	none	PD	ND	N	N	N	ND
**253**	F	51	Y	N	9.2	FGFR2-TACC2	PD	Neg	N	N	N	Neg
**276**	F	50	Y	Y	3.7	none	PD	Pos	Y	N	N	ND
**305**	M	67	Y	Y	6.7	none	PD	Pos	N	N	N	Neg
**338**	F	77	Y	Y	11.6	none	PD	Pos	N	N	N	Neg
**348**	M	57	Y	Y	4.1	none	PD	ND	N	N	N	Neg
**357**	F	37	Y	N	105.5	none	PD	Pos	N	N	N	Neg
**359**	M	66	Y	N	4.5	none	PD	Pos	N	N	N	Neg
**14**	M	74	N	Y	NA	FGFR3 amplification, FGFR3-FAM175B	PD	Pos	Y	N	N	Neg
**211**	F	52	N	Y	NA	FGFR1 amplification	PD	Pos	N	N	N	ND
**233**	M	59	N	N	NA	FGFR1 amplification, FGFR1-RTN4 fusion	PD	ND	N	N	N	ND
**254**	M	76	N	Y	NA	FGFR2	WD	Pos	N	N	N	Neg
**267**	M	77	N	N	NA	FGFR1 amplification	WD	ND	Y	Y	N	Neg
**339**	M	81	N	N	NA	FGFR4 Amplification	PD	Pos	N	N	N	Neg

FGFR: fibroblast growth factor receptor; ERBB2: Erb-B2 receptor tyrosine kinase 2; TMB: tumor mutational burden; MSI: microsatellite instability; EBV: Epstein–Barr Virus; PD-L1: programmed death-ligand 1; CPS: combined positive score; F: female; M: male; Y: yes, N: no; NA: not applicable; PD: poorly differentiated; MD: moderately differentiated; WD: well-differentiated; Neg: negative; Pos: positive; ND: not done.

## Data Availability

Not applicable.

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
