# Peer review of "Peritoneal Seeding Is More Common in Gastric Cancer Patients with FGFR2 Amplification or High Tumor Mutation Burden"

_diagnostics, 2022, doi:10.3390/diagnostics12102355_

Round 1
Reviewer 1 Report
I thank the authors for the particular point of view in treating the carcinogenesis due to gastric cancer.
The paper is well structured and tries to describe as well as possible the molecular mechanisms and cell mediators in the process of carcinogenesis.
It is very important nowadays, with the development of the immune therapy, the compression and the discovery of the complex pathways of carcinogenesis.
It would be very interesting if the authors can integrate in this article data about the chemotherapeutic treatment of the patients with FGFR2 amplification and their follow-up and survival
Author Response
We greatly appreciate your valuable efforts to carefully review the manuscript and the constructive suggestions offered. A revision of the manuscript has been carried out to take all of them into account. We believe the manuscript has been improved and is suitable for consideration for the publication. Each comment is followed by the corresponding response highlighted by blue. In addition, we made corrections and clarifications as the reviewer suggested, all of them was highlighted and underlined in the final manuscript.
Comment: I thank the authors for the particular point of view in treating the carcinogenesis due to gastric cancer.
The paper is well structured and tries to describe as well as possible the molecular mechanisms and cell mediators in the process of carcinogenesis.
It is very important nowadays, with the development of the immune therapy, the compression and the discovery of the complex pathways of carcinogenesis.
It would be very interesting if the authors can integrate in this article data about the chemotherapeutic treatment of the patients with FGFR2 amplification and their follow-up and survival.
Response: We sincerely appreciate your positive comments and suggestion. We have added the chemotherapeutic regimens (page 4, highlighted and underlined) and discussion on the chemotherapeutic treatment of the FGFR2-amplified GC patients (page 6, highlighted and underlined) as below.
For the treatment of FGFR2-amplified GC patients, adjuvant or palliative chemotherapy was administered to 25 cases because one patient had refused chemotherapy. Specific regimen includes xelox (capecitabine and oxaliplatin) in all but one of the cases as the first-line chemotherapy. For patients with progress of disease after xelox, paclitaxel (taxol) and ramucirumab (n=12), folfox (oxaliplatin-5-fluorouracil-leucovorin) (n=5) or folfiri (leucovorin calcium, fluorouracil, and irinotecan) (n=2) combination chemotherapies, and pembrolizumab (n=5) were used as the second- or third-line chemotherapy. In one patient, Derazantini, a potent FGFR1‒3 kinase inhibitor was administered.
There have been several studies that showed worse survival of FGFR2 amplified or overexpressed GCs. [33, 34] Although selective FGFR2 inhibitor has shown to be more sensitive to FGFR2 amplified GC cell-lines, there are four ongoing clinical trials (NCT01719549, NCT01921673, NCT02052778, NCT02318329) to assess the efficacy in actual patients. [22]
Reviewer 2 Report
The study focuses on FGFR2 amplification and highlights FGFR2 as a potential marker for peritoneal seeding of gastric carcinogenesis. However, this study lacks some innovative perspective, since FGFR2 amplification in GC is a well-known mechanism and even in many other GC subtypes. I am impressed by the author's study design and data interpretation. However, the novelty of the current manuscript could be improved.
Author Response
We greatly appreciate your valuable efforts to carefully review the manuscript and the constructive suggestions offered. A revision of the manuscript has been carried out to take all of them into account. We believe the manuscript has been improved and is suitable for consideration for the publication. Each comment is followed by the corresponding response highlighted by blue. In addition, we made corrections and clarifications as the reviewer suggested, all of them was highlighted and underlined in the final manuscript.
Comment: The study focuses on FGFR2 amplification and highlights FGFR2 as a potential marker for peritoneal seeding of gastric carcinogenesis. However, this study lacks some innovative perspective, since FGFR2 amplification in GC is a well-known mechanism and even in many other GC subtypes. I am impressed by the author's study design and data interpretation. However, the novelty of the current manuscript could be improved.
Response: Thank you for your critical comments and suggestion. It is true that FGFR2 amplification is well known in GC; however, the novelty of our study was that FGFR2 amplification was found to be associated with peritoneal seeding. We have emphasized our main topic on the end of our discussion (last paragraph of page 6 to first paragraph of page 7, highlighted and underlined) as below.
Although there have been previous studies to report FGFR2 amplification in gastric cancers and its association with poor outcome, our study has shown for the first time that FGFR2 amplification is significantly associated with peritoneal seeding. Further study with larger sample size is warranted to confirm our results. Moreover, a prospective study on FGFR-altered GC cases with respect to targeted drug response would be valuable.